# The Enzyme Gene Expression of Protein Utilization and Metabolism by *Lactobacillus helveticus* CICC 22171

**DOI:** 10.3390/microorganisms10091724

**Published:** 2022-08-26

**Authors:** Huixin Zhang, Mengfan Xu, Shanhu Hu, Hongfei Zhao, Bolin Zhang

**Affiliations:** Beijing Key Laboratory of Forest Food Processing and Safety, College of Biological Science & Biotechnology, Beijing Forestry University, Beijing 100083, China

**Keywords:** *Lactobacillus helveticus*, β-casein, hydrolysis capacity, hydrolysis system

## Abstract

The purpose of this study was to explore the hydrolytic ability of *Lactobacillus helveticus* CICC 22171 with regard to protein and the expression of enzyme genes during protein utilization. The results revealed that the strain hydrolyzed casein from the C-terminal, reached the maximum level in 6 h, and the number of amino acids in the hydrolyzed peptide was 7–33. The molecular weight was 652.4–3432.74 kDa. Hydrophobic peptides produced by hydrolysis were the source of β-casein bitterness. Leucine and glutamine were the preferred cleavage points after 1 h; tyrosine and tryptophan subsequently increased. The first step of hydrolysis was controlled by PrtP and PrtM genes and coordinated with the action of PrtH1 and PrtH2. The transport system consisted of DtpT, OppB, OppD and OppF. The hydrolytic third step endopeptidase system consisted of the aminopeptidases (PepN, PepC, PepM and PepA), the endopeptidases (PepE, PepF and PepO); the dipeptidases (PepV and PepD), the tripeptidase PepT; the proline peptidases (PepX, PepP, PepQ, PepR and PepI). The expression of CEP genes was significantly different, and the expression level of genes related to the transport system significantly increased from 0 to 1 h. The specificity of the substrate and action site of endopeptidase was abundant.

## 1. Introduction

Lactic acid bacteria (LAB) are important strains used in the food industry, which are widely added in the production of pickled and fermented foods, such as cheese, yogurt, wine, and kimchi [1]. As probiotics, it has a variety of physiological functions on the human body, such as cholesterol lowering function, immune regulation, antioxidant properties, etc. [2]. During the long evolutionary process, lactic acid bacteria have developed an efficient protein hydrolysis system. At present, there are a large number of studies on the protein hydrolysis system of *Lactococcus lactis* [3]. The structural characteristics of the proteolytic system of *Lactobacillus* are similar to that of *L**c. lactis*. For example, the proteolytic system of *Lc. lactis* MG 1363 and *L**c. lactis* IL 1403 include peptidase and the transport system [1,4]. In this system, protein hydrolysis mainly consists of the following three steps: in the first step, protein is hydrolyzed into small molecular polypeptides. In the second step, peptides are transported into cells through the transport system. Third, after entering the cell, the polypeptide is further hydrolyzed by peptidases into free amino acids [1]. Among them, the extracellular enzymes are a series of cell envelope proteinases (CEP), which mainly hydrolyze exogenous proteins, such as β-casein, into polypeptides, which are then transferred into the cell by the polypeptide transport system, and hydrolyzed into free amino acids under the action of intracellular peptidases in the cell for the growth of the bacteria. Among them, there are many kinds of intracellular enzymes, mainly including endopeptidase and exopeptidase, which can be subdivided into aminopeptidase, carboxypeptidase, dipeptidase and tripeptidase, according to the differences in action sites [5].

The mass concentration of nitrogen in cell dry weight is second only to carbon and oxygen, and it is an important element in the composition of nucleic acid and protein, and plays an important role in the growth of microorganisms [6]. In the common culture medium of lactic acid bacteria, the nitrogen source is usually peptone, beef extract and yeast powder. The nitrogen and carbon sources in the recovered skim milk medium are milk proteins, mainly casein. Studies have shown that for *Lactobacillus helveticus* DPC4571 in the same growth period, compared with the basic MRS medium, the polypeptidyase activity of the bacteria in the recovered skim milk medium significantly increased, with the difference of 2–6 times in the logarithmic phase and 2 times in the stationary phase. This difference in medium composition can lead to the up-regulation of peptide enzyme synthesis in skim milk media. Casein can be divided into the following four types: α_s_-CN, β-CN, γ-CN and κ-CN. The structural characteristics of casein make it difficult for the human body to digest it and it is prone to causing allergic reactions. In recent years, many peptides with important physiological functions have been found in casein hydrolysates, such as the opioid peptide, blood pressure lowering peptide, antithrombotic peptide, immune promoting peptide, mineral ion absorption promoting peptide, etc. [7]. These peptides are about 1000 Da and are easily absorbed by the intestine. This makes the hydrolysis of casein a research hotspot [6].

*Lb. helveticus* has a large number of physiological activities. On the one hand, through living bacteria cells, it can regulate the human intestinal tract, maintain the balance of intestinal flora; on the other hand, the active components obtained by the metabolism of bacteria, especially the polypeptides obtained by fermentation, such as the angiotensin converting enzyme (ACE) inhibitory peptide, can effectively regulate human function, improve sleep and reduce blood pressure [8,9]. In addition, *Lb. helveticus* has been added to cheese and dairy products to improve their flavor [10]. The strain CICC 22171 used in this study has been proved to have potent probiotic functions. However, only glycometabolism and energy metabolism have been studied in this strain, and its specific protein utilization is unknown [11]. In addition, there are no data to support the gene regulation of the transport system of strain CICC 22171. Therefore, this study further explored the proteolytic ability and enzyme gene expression of strain CICC 22171. The metabolic types of LAB are diverse, and there are differences among species and genera. The utilization process of protein by *Lc. lactis* is clear, while for *Lb. helveticus*, it is still insufficient. Therefore, the purposes of this study were as follows: (1) to observe the ability of *Lb. helveticus* CICC 22171 to produce free amino acids and short peptides by hydrolyzing β-casein; (2) analyze the composition of the *Lb. helveticus* CICC 22171 protease hydrolysis system; (3) determine the gene expression during *Lb. helveticus* protease hydrolysis.

## 2. Materials and Methods

### 2.1. Microorganisms and Culture Conditions

*Lb. helveticus* CICC 22171 was isolated from traditional Chinese cheese, and preserved in the China center of industrial culture collection (CICC). The media used in this study were De Man, Rogosa, and Sharpe (MRS) media; the frozen strains were activated in MRS medium and subcultured at 5% inoculation volume. The cells were cultured at 37 °C until the cell concentration was 5 × 10^9^ cfu/mL, and the bacteria were collected by centrifugation (4 °C, 6000 r/min, 10 min). 

### 2.2. Hydrolysis of β-Casein

β-casein (Sigma-Aldrich, St. Louis, MO, USA) was used as the substrate to prepare a solution of 15 mg/mL. The bacterial suspensions were mixed with equal volumes of solution. The mixture was cultured at 35 °C, and samples were collected at 0 h, 1 h, 3 h, 5 h, and 6 h, respectively. The supernatant was collected by centrifugation at 10,000 r/min for 5 min and filtered by a 0.45-μm microporous membrane. The filtrate was stored at −20 °C for further analysis.

### 2.3. Determination of Hydrolytic Capacity

The proteolysis capacity was measured by electrophoresis apparatus (Bio-Rad, Hercules, CA, USA). Sodium dodecyl sulfate-polyacrylamide gel electrophoresis (SDS-PAGE) was employed with 5% stacking gel and 15% separating gel. The hydrolysis solution was mixed with the sample buffer. Then, 7 µL of each sample and the standard protein were added into each sample tank. The initial voltage was 100 V, and was adjusted to 120 V after the sample entered the separation glue, so that the blue bottom edge of the sample was close to the bottom of the glass plate 1–2 cm. After migration, the gel was stained in 0.25% Coomassie blue R-250 solution and decolorized with a mixture of 30% methanol and 10% acetic acid.

### 2.4. Identification and Analysis of Hydrolysates

The peptide composition of the β-casein hydrolysate was analyzed by high performance liquid chromatography–electrospray tandem mass spectrometry (LC–ESI MS/MS), and the supernatant hydrolysates were obtained at 1, 3 and 5 h. Capillary high performance liquid chromatography (Capillary HPLC) was performed on a C-18 liquid chromatography column (Shimadzu, Kyoto, Japan). The elution gradient parameters were set to 0.1% aqueous formic acid solution (mobile phase A); acetonitrile acidified with 0.1% formic acid (mobile phase B); the elution gradients were 95% A (0–16 min), 90% A (16–51 min), 78% A (51–71 min), 70% A (71–72 min), 5% A (72–78 min), with a flow of 600 nL/min.

The enzymatic hydrolysates were separated and desalted by capillary HPLC (LC-20AT, Shimadzu, Kyoto, Japan). The analysis was performed using a Q Exactive mass spectrometer (Thermo Fisher, Walthamm, MA, USA) with positive ion detection. The peptides were collected and the mass-to-charge ratios of their fragments were determined.

### 2.5. Exploration of the Gene of Proteolytic Enzymes

In this study, the composition of the relevant enzymes in the proteolytic system of *Lb. helveticus* was explored by PCR. DNA of the bacteria was extracted with a bacterial genomic DNA rapid extraction kit (Sangon Biotech Shanghai Co., Ltd., Shanghai, China). The extracted DNA was subjected to agarose gel electrophoresis to detect the quality. The primers were designed and synthesized using Primier 5.0 software (Premier biosoft, CA, USA). All the primers involved are shown in Appendix A. The corresponding genes were amplified according to the 50 μL PCR system. The amplification was performed at 95 °C for 10 min, followed by 30 cycles of 94 °C for 30 s, 52 °C for 1 min, and 72 °C for 90 s and extended to 10 min at 72 °C. 

### 2.6. Analysis of Gene Expression of Proteolytic Enzymes

The gene expression changes in the proteolytic enzymes for different hydrolysis times were determined by real-time PCR. The RNA of the thallus was extracted by Trizonl (Sangon Biotech Co., Ltd., Shanghai, China). The purity of RNA was tested using a nanophotometer spectrophotometer (Implen, Westlake Village, CA, USA). Reverse transcription was performed as follows: a sterilized Eppendorf tube without RNA enzyme was taken, and each sample was added to 12 µL of Mix I; Mix I was placed in a 65 °C warm water bath for 5 min, and then ice was immediately added for 1 min; 20 µL Mix II consisted of 4 µL of 5× first-strand buffer, 2 µL of 0.1 M dTT, 1 µL of RNaseOUT 40 U/µL, 1 µL of SuperScrip III RT (200 U/µL) and 12 µL of Mix I. The treatment conditions were as follows: 25 °C for 5 min, 42 °C for 60 min, 70 °C for 15 min. It was placed on ice immediately. The obtained cDNA can be stored at −20 °C for half a year.

The genes were selected according to the PCR results, and the primers were designed and synthesized by the Primier 5.0 software. The primers involved are shown in Appendix A. Real-time PCR was used to analyze its expression. The volume of each component was 2 µL of 10× PCR buffer, 1 µL of Mg^2+^ (50 mM), 0.5 µL of dNTPs (10 mM), 0.5 µL of forward primer (10 µM), 0.3 µL of SYBR (20×), 0.5 µL of reverse primer (10 µM), 0.2 µL of platinum taq polymerase and 1.0 µL of the template. RT-PCR was performed under the following conditions: 1 cycle of holding stage at 95 °C for 20 s; 40 cycles of cycling stage at 95 °C for 3 s.

### 2.7. Statistics

All experiments and analyses were repeated for 3 times, and the data represented the mean value. Microsoft Office Excel 2010 and SPSS 18.0 software (SPSS Inc., Chicago, IL, USA) were used for data analysis, and Origin 2021 software was used for drawing the figures.

## 3. Results

### 3.1. Determination of Protease Hydrolysis Capacity

Figure 1a reveals the hydrolysis profile of β-casein caused by *Lb. helveticus* CICC 22171 in co-cultivation at different times. The main band of standard β-casein ranged from 30 to 20 kDa. With the extension of the hydrolysis time, the main band of electrophoresis became thinner and the bands with smaller molecular weight increased. The results indicated that *Lb. helveticus* had a good ability to hydrolyze protein, and the degree of hydrolysis was positively correlated with time.

### 3.2. Identification and Analysis of Hydrolysates

The data of the mass spectrometric analysis of peptides are shown in Table 1; the number of amino acids of the peptide fragments obtained by hydrolysis ranged from 7 to 33. Among the 1 h hydrolysate products, there were 103 peptides with molecular weights that ranged from 787.40 to 4928.75, and 8 of them were modified by oxidation. The hydrolysate contained 424 peptides at 3 h, with molecular weights that ranged from 652.40 to 4928.74, and 84 of them were oxidized. The hydrolysate contained 428 peptides at 5 h, with molecular weights that ranged from 652.4 to 3432.74, and 85 of them were oxidized. The amino acids modified by oxidation were all methionine (M). The results showed that the longer the co-culture time was, the more the number and types of the peptides increased, and the wider the molecular weight range was. According to the hydrolysate products at 1 h (Figure 1b), it can be observed that the obtained peptides were mainly concentrated at the C-terminus of the hydrophobic sequence, which indicated that the hydrolysis of β-casein started from the C-terminus. *YQEPVLGPVRGPFPIIV* was a bitter peptide sequence. In this study, the peptides included in this sequence obtained by hydrolysis are shown in Table 2. The hydrolysate only contained the same fragment as the studied sequence at 5 h, while the sequences at 3 h and 1 h also contained other amino acids that had not been hydrolyzed away. These data indicated that with the increase in hydrolysis degree, the spatial conformation gradually opened, and more hydrophobic amino acid sites were exposed.

The cleavage points of peptides in hydrolysates at 1, 3 and 5 h were analyzed by combining them with the standard sequence of β-casein. By comparing 1 h and the overall level, the positions of β-casein hydrolyzed by bacteria were mainly behind the following 19 amino acids, and the number distribution is shown in the Figure 1c. The results revealed that the preferred cut-point of the bacteria was mainly behind the aliphatic amino acids, mainly leucine (L) and glutamine (Q). From the cut-off point at 1 h, more hydrophobic amino acid cut-off points appeared with the extension of culture time, especially aromatic amino acids, such as tyrosine (Y), tryptophan (W), etc. At the same time, more and more fragments were obtained by hydrolysis when the cut point was at the aliphatic amino acids. The least active sites were hydrophilic amino acid glycine (G). On the other hand, it can be concluded that the tyrosine (Y), tryptophan (W), aspartic acid (D) and arginine (R) in the β-casein sequence were all hydrolyzed by the bacteria. In the end, it was inferred that the main action point of the cell wall anchoring the proteases of *Lb. helveticus* lies in these four amino acids.

### 3.3. Analysis of Related Enzyme Composition of Proteolytic System

The gene amplification results of the proteolytic system are shown in Table 3. A series of genes associated with CEPs included PrtH, PrtP, and PrtM. A series of genes associated with bacterial peptide transport systems included OPPB (transmembrane protein), OPPD, OPPF (nucleotide binding protein) and DtpT (proton-powered dipeptide and tripeptide carriers). The hydrolytic third step endopeptidase system consisted of the aminopeptidases PepN, PepC, PepM and PepA, the endopeptidases PepE, PepF and PepO; the dipeptidases PepV and PepD, the tripeptidase PepT; the proline peptidases PepX, PepP, PepQ, PepR and PepI.

### 3.4. Analysis of CEP Gene Expression

The expression of CEPs is shown in Figure 2a. With the increase in hydrolysis degree, the changes in gene expression were significantly different. The expression levels of PrtP2 and PrtH2 increased at first, significantly increased and reached the maximum at 1 h, and then decreased. The expression level of PrtM always showed a downward trend, which was significantly lower than that of 0 h. The expression of PrtH1 was the highest at 6 h.

### 3.5. Analysis of Polypeptide Transporter Gene Expression

The expression of a series of genes related to the bacterial polypeptide transport system is shown in Figure 2b. The expression of DtpT in the polypeptide transport system increased significantly at 1 h, and the three genes (OPPB, OPPD and OPPF) showed similar changes from 0 to 3 h. Except for OPPB, the other two genes OPPD and OPPF showed an upward trend from 3 to 6 h. Combined with the fact that the specific substrates for each gene were different, it can be concluded that a large number of dipeptides and tripeptides were produced during the 0–1 h period, and the corresponding transport enzymes were needed to transport them into the cell. Based on the expression levels of all transport-related enzymes, the expression levels were generally up-regulated. Therefore, the strains were constantly hydrolyzing milk protein to produce peptides and amino acids in the co-culture process.

### 3.6. Analysis of Endopeptidase Gene Expression

#### 3.6.1. Analysis of Aminopeptidase Expression

The changing trend of aminopeptidase expression is shown in Figure 2c. The results revealed that only PepM decreased and then increased, and the expression of gene was significantly different from each other in all four periods. The expression levels of PepN1 and PepC were significantly up-regulated from 0 to 1 h, but showed a downward trend during the period of 1–6 h. However, PepN2 and PepA increased significantly at 1 h and then decreased to the lowest value at 3 h, and showed an upward trend during the period of 3–6 h. It was worth noting that the variation trend of PepN1 and PepN2 genes that encode PepN was inconsistent.

#### 3.6.2. Endopeptidase

The expression of endopeptidase is shown in Figure 2d. The results revealed that the expression levels of each gene were significantly different in all four periods and increased significantly at 1 h. The three genes of PepE showed a downward trend after 1 h, and the expression levels at the end of culture were all lower than those at the beginning of culture. The expression of PepF decreased at 3 h, but increased at 6 h. PepO1 reached its maximum value at 3 h. The variation trend of PepO2 and PepO3 was similar; both significantly increased at 1 h and then decreased, but the range of expression was different.

#### 3.6.3. Dipeptidase and Tripeptidase

The expression trend results of dipeptidase and tripeptidase are shown in Figure 2e. In dipeptidase, the expression levels of PepV and PepD were completely different. The fluctuation range of PepV was smaller than PepD, and the maximum expression level appeared at 6 h, which was 1.3 times more than the initial expression level. The five genes encoding PepD showed different trends. PepD5 and PepD4 showed the same trend; both increased significantly from 0 to 1 h, decreased from 1 to 3 h, and increased from 3 to 6 h. However, PepD1, PepD2 and PepD3 showed the same trend and decreased after 1 h, among which PepD2 and PepD3 decreased significantly. Meanwhile, PepD3 fluctuated the most, but the expression levels of all five genes reached their highest expression levels in 1 h. The two genes encoding tripeptidase had the same change trend and both reached the maximum value at 1 h.

#### 3.6.4. Proline Peptidase

The expression trend of proline peptidase is shown in Figure 2f. The results indicated that PepI and PepR had the same trend among the many proline peptidases detected, both of which increased significantly from 0 to 1 h. PepP increased slightly at 1 h, but with the increase in hydrolysis time, the expression of PepP and PepQ decreased gradually. The expression levels of PepP were significantly different at different periods, while PepQ was significantly decreased at 3 h. On the contrary, the expression level of PepX increased gradually with the prolongation of culture time, and reached the maximum at 6 h. On the contrary, with the prolongation of culture time, the expression level of PepX gradually increased, and there were significant differences in each period, and reached the maximum at 6 h.

## 4. Discussion

Due to the lack of free amino acids and low level of molecular peptides that are essential for the growth of *Lb. helveticus* in milk, the bacterium required an active protease system in order to hydrolyze the protein in milk to amino acids and peptides [12]. In this study, the electrophoretic bands narrowed with the prolongation of hydrolysis time. *Lb. helveticus* CICC 22171 was able to produce short peptides and release amino acids from the casein matrix. Therefore, it had certain proteolytic activity. In the study of *Lactobacillus delbrueckii* subsp. *bulgaricus* 92059 degradation of casein, the results showed that β-casein had the best degradation effect [13]. In the previous study, the result of SDS-PAGE revealed that the bands of *Lactobacillus delbrueckii* subsp. *lactis* ACA-DC 178 narrowed when hydrolyzing β-casein and also demonstrated good hydrolysis ability [14]. In another study, it was also confirmed that a large amount of casein was hydrolyzed during the first hour of incubation. Some studies obtained 20 peptides, with an average length of 14.1 amino acid residues by hydrolysis of β-casein [15].

Hydrophobic peptides were associated with a bitter taste. During the hydrolysis of casein, a large number of hydrophobic amino acid polypeptides were produced, which became the main source of casein bitterness [16]. In general, the bitterness was most obvious when hydrophobic amino acids were located in the middle of the peptide chain, followed by those located in the two ends, and the free amino acids were the weakest. However, proteases that specifically acted on hydrophobic amino acids were more conducive to hydrolyzing peptides whose terminal residues were hydrophobic amino acids. As the degree of hydrolysis increased, bitter amino acids, such as lysine, leucine and methionine, were gradually exposed from the middle of the peptide chain, and the bitterness decreased. The proteolytic enzyme system of *Lb. helveticus* CNRZ 32 has been studied the most thoroughly [3,17,18]. *Lb. helveticus* CNRZ 32 could also reduce bitterness and accelerate the development of cheese flavor [3,12]. In the study of protein hydrolysis of *Lc. lactis* ssp. to produce cheese, it was found that *Lc. lactis* ssp hydrolyzed more hydrophobic peptides and the content of free amino acids increased with the prolongation of hydrolysis time; it indicated that different species had similarities in the reduction in bitter peptides in protein hydrolysis [19]. Cheese produced by *Lc. lactis* subsp. *cremoris* SK11 had a much lower intensity of bitterness than that produced by a commercial starter [20]. These results could be used to further study the bitter removal of casein in milk.

In this study, the results indicated that the strains have many hydrolysis sites for β-casein and are not limited to a few amino acids. In the study of 15 different *Lb. helveticus* CEPs, it was found that the strain had a rich distribution of hydrolytic sites, including aliphatic amino acids, such as leucine, valine, alanine and isoleucine, dicarboxylic amino acids, such as glutamic acid and glutamine, and basic amino acids, such as lysine and arginine, aromatic amino acids, such as phenylalanine, tyrosine and tryptophan, hydroxylated amino acids, such as serine and threonine, sulfur-containing amino acids, methionine and heterocyclic amino acids, such as proline, and the hydrolysis sites of β-casein showed great differences between the different strains. Partial cut points were found only in the strains for which both proteases of strain ITGLH1 and strain roselle 5088 existed. It was worth noting that no matter how many proteases were present in the strain, two-thirds of the cut point occurred at the C-terminus of the sequence [21], which was consistent with the results of this study. For example, *Lb. helveticus* CNRZ 303 and CP790 acted on most of the chemical bonds at the C-terminus of β-casein, while CNRZ 32, CNRZ 303 and LHC2 were closer to the N-terminal, and the product characteristics of the different strains lied in the casein co-culture phase; *Lb. helveticus* CNRZ 32 hydrolyzed the most diverse peptides [22]. Interestingly, in a study related to the cleavage points of *Lc. lactis*, *Lc. lactis* subsp. *cremoris* WG2 hydrolyzed the C-terminus of the β-casein, but the N-terminus of the molecule was poorly hydrolyzed, and 91 different cleavage points were detected, indicating that *Lb. helveticus* had the same cleavage point preference as *Lc. Lactis* [23].

In previous studies, whole genome sequencing (WGS) and comparative analysis of protein components of *Lb. helveticus* LH-2 and *L. acidophilus* LA-5 proteolytic systems were performed. The results showed that there were some genetic differences in the distribution of hydrolytic components between LH-2 and LA-5 [24]. It has been reported that *Lb. helveticus* strains may have one or more genes that encode CEPs, which could hydrolyze casein into oligopeptides. Only the PrtH gene was present in *Lb. helveticus* BGRA 43, which had 98.9% homology with the same gene of *Lb. helveticus* CNRZ32, which was consistent with the results of this study. In *Lb. helveticus*, activation of CEPs requires a mature protease called PrtM. Two PrtMs (PrtM1 and PrtM2) were found in *Lb. helveticus* CNRZ32 and only one in this study [12]. *Lb. helveticus* DPC4571 had three peptide transport systems, which were the oligopeptide transport system and dipeptide and tripeptide transport system, Dpp and DtpT. In contrast, H10 had two peptide transport systems, Opp and DtpT, and these results suggested that there were differences in the proteolytic systems among the different strains [25,26]. The *L**b. helveticus* proteolytic system had been proven to contain the following eleven enzymes: two proline-specific endopeptidases PepE and PepO; one tripeptidase PepT; four aminopeptidases, PepX, PepI, PepQ and PepR; four dipeptidases, PepD, PepV, PepC and PepN, which was similar to the results of this study [27,28]. In the same way, the protein system of *L**c. lactis* MG 1363 consisted of twelve peptidases (PepC, PepN, PepX, PepP, PepA, PepF2, PepDA1, PepDA2, PepQ, PepT, PepM and PepO1), proteases (PrtP1 and PrtP3) and three transport systems (DtpT, DtpP and OPP-PEPO1) [1]. The potential peptidases PepDA1 and PepDA2 were found in the *Lc. lactis* IL 1403 gene, which were similar in sequence to the PepD dipeptidase of *Lb. helveticus* and PepM, suggesting some similarity between *Lc. Lactis* and *L**b. helveticus* [4].

PrtP was a serine protease on the cell wall. As a protease on the cell membrane, PrtM can encode membrane-associated lipoproteins, which is closely related to the maturation and catalysis of the protease on the cell wall surface [12]. The first step in using milk proteins is to degrade these proteins to oligopeptides by extracellular proteases (PrtP). The proteases were capable of hydrolyzing β-casein into more than 100 different oligopeptides, ranging from 4 amino acid residues to at least 30 amino acid residues [23]. PrtP was responsible for transcription into the precursor protein of CEPs. At the beginning of co-cultivation, a large amount of protease was needed to be synthesized to hydrolyze casein. Therefore, the expression level of PrtP was the highest at the initial stage, and then gradually decreased with the extension of the culture time.

CEPs of *Lb. helveticus* had great intraspecies differences. Unlike other types of proteases in lactic acid bacteria, PrtH1 and PrtH2 in *Lb. helveticus* lacked the anchoring domain at the C-terminus of the CEP’s sequence. Some scholars believe that there are at least two other proteases [21,29,30]. In the comparative study of PrtH1 and PrtH2, it was found that 29 strains of *Lb. helveticus* could be detected for PrtH2, and PrtH was only found in 18 of them. This indicated that compared with other protease genes, PrtH2 was ubiquitous in *Lb. helveticus* [31]. The results of this study confirmed this phenomenon by the discovery of the PrtH2 gene in the strain CICC 22171. It had been reported that PrtH was not widely distributed in *Lb. helveticus* strains, among which PrtH3 has the strongest homology, and more than 80% of the strains were homologous to PrtH3. PrtH2 showed poor homology. The distribution of specific genes was also related to the strains [31]. The combination of PrtH3/PrtH4 was common in *Lb. helveticus* [32]. For example, the genes of PrtH3 and PrtH4 were found in the genome of *Lb. helveticus* LH-2, while PrtH and PrtH2 were absent, which was different from the results of this study [24]. PrtM was not present in LH-2, whereas it was present in this study, indicating that PrtM may be related to PrtH [26].

For *Lc. lactis*, the complete oligopeptide transport system included OPPB, OPPC (transmembrane hydrophobic region), OPPD, OPPF (ATP-binding domain) and OPPA (specific ligand binding protein) [33]. However, only OPPB, OPPD and OPPF were obtained in this study. The results showed that OPPB and OPPC could act synergistically in the oligopeptide transport system of the strain CICC 22171, and the transport system could function normally, even when one of the genes was deleted.

Aminopeptidase could cut off the hydrophobic amino acid in the bitter peptide from the N-terminus of the sequence, so it could remove the bitterness [34]. In addition, different kinds of aminopeptidases had different hydrolysis specificities for amino terminal residues. Among them, lysyl aminopeptidase PepN and phenylalanyl aminopeptidase PepM had low specificity, which could hydrolyze most amino acid residues. Similarly, PepC had poor specificity, which could act on oligopeptides with 2–12 amino acids, while PepA could specifically act on glutamine and asparagine residues [35].

PepO2 was associated with the hydrolysis of casein-derived bitter peptides [36]. In this study, PepO increased significantly from 0 to 1 h, which indicated that a large number of bitter peptides were hydrolyzed at this stage. The similarity of changes in PepO2 and PepO3 expressed that the two genes had similar functions in the modified strain, but some differences existed. In research on the hydrolysis of bitter peptides from different casein sources of *Lb. helveticus* CNRZ32, PepO3 and PepO2 had similar functions, while PepE and PepF had certain differences and the results were similar to this study [37].

Based on the above data analysis, the peptidase system was composed of peptidase PepN /M/C/A, proline peptidase PepP/X/I/R/Q, dipeptidase PepV/D, tripeptidase PepT and peptidase Pepe/F/O, and part of the enzymes had the coordination of the corresponding coding genes [38]. According to the analysis of the expression level, a large number of oligopeptides that contained proline and bitter peptides were produced in the thalli within 0–1 h, which were further decomposed after being transported into the cell. Moreover, due to the diversity of hydrolysates, the expression of peptidases was also significantly different. Recent research identified 16 peptidase genes in the genomic DNA of 50 sequenced strains of *Lb. helveticus*, which were isolated from various environments, including fermented dairy products, non-dairy products, and human feces. Aminopeptidases that belong to the superfamilies PepC and PepN and proline peptidases PepX were present in all genomes, usually with one gene, while proline peptidases PepL were absent in all the strains [39].

## 5. Conclusions

The degree of β-casein hydrolysis of *Lb. helveticus* CICC 22171 was positively correlated with hydrolysis time. The number and type of peptides increased with the extension of co-culture time, and the range of molecular weight increased. *Lb. helveticus* CICC 22171’s preferred cleavage point was mainly in aliphatic amino acids, and hydrophobic amino acids in casein hydrolysates were the main source of casein bitterness. A total of 22 genes related to *Lb. helveticus* CICC 22171 that hydrolyzed β-casein were identified. CEPs of *Lb. helveticus* had great intraspecies differences. For strain CICC 22171, the CEPs had both PrtH1 and PrtH2. In the transport system, only OppB, OppD and OppF were obtained in the strain. The endopeptidase system consisted of PepN/M/C/A, PepP/X/I/R/Q, PepV/D, PepT and PepE/F/O. Some enzymes demonstrated coordinated action of multiple encoding genes, such as PepN and PepO, and their expression trends were different. Due to the diversity of hydrolysates, the expression of peptidases was different. The two genes had similar functions but different expression levels. The variation trend of endopeptidase in different species was significantly different.

## Figures and Tables

**Figure 1 microorganisms-10-01724-f001:**
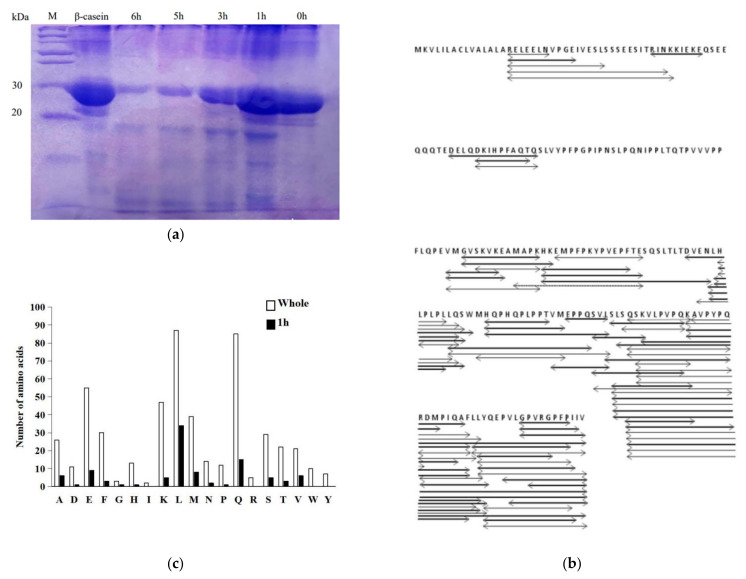
(**a**) shows the hydrolysis profile of β-casein caused by *Lb. helveticus* CICC 22171, (**b**) represents the peptides contained after 1 h, (**c**) shows the numbers of the cleavage sites after 1 h and for the whole time.

**Figure 2 microorganisms-10-01724-f002:**
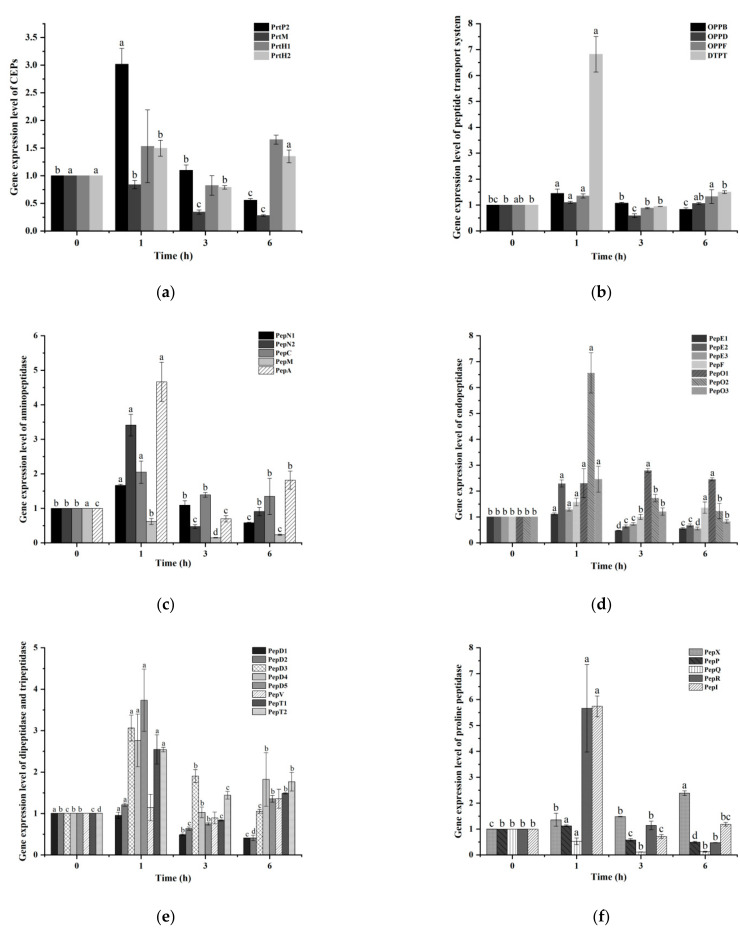
Gene expression of different enzymes. (**a**) represents the expression level of CEP-related genes, (**b**) reveals the level of gene expression of the peptide transport system, (**c**) describes the gene expression level of aminopeptidase, (**d**) is the gene expression level of endopeptidase, (**e**) shows the gene expression level of dipeptidase and tripeptidase, (**f**) explains the gene expression level of the related proline peptidase. Different letters indicate significant differences (*p*-value of ≤0.05) within each subsection of the figure.

**Table 1 microorganisms-10-01724-t001:** LC gradient parameters.

Incubation Time	1 h	3 h	5 h
Number of hydrolyzate peptides/piece	103	424	428
Number of hydrolyzed peptides modified by oxidation/piece	8	84	85
The range of hydrolysate peptide molecular weight /Da	787.40–4928.75	652.40–4928.74	652.4–3432.74

**Table 2 microorganisms-10-01724-t002:** Bitter peptide of hydrolysate.

Sequence	MH + (Da)	1 h IonScore	3 h IonScore	5 h IonScore
YQEPVLGPVRGPFPIIV	1881.06	26.61	36.76	39.80
LYQEPVLGPVRGPFPIIV	1994.15	16.62	31.51	31.81
LLYQEPVLGPVRGPFPIIV	2107.23	29.21	37.12	42.84
KVLPVPQKAVPYPQRDMPIQAFLLYQEPVLGPVRGPFPIIV	4610.58	1.32	-	-
SKVLPVPQKAVPYPQRDMPIQAFLLYQEPVLGPVRGPFPIIV	4697.59	15.78		
QSKVLPVPQKAVPYPQRDMPIQAFLLYQEPVLGPVRGPFPIIV	4825.68	19.14		
SQSKVLPVPQKAVPYPQRDMPIQAFLLYQEPVLGPVRGPFPIIV	4912.73	1.79	14.35	
SQSKVLPVPQKAVPYPQRDM*PIQAFLLYQEPVLGPVRGPFPIIV	4928.75	9.62	12.47	

**Table 3 microorganisms-10-01724-t003:** Gene amplification results of the proteolysis system.

Number	Amplification Results	Similarity
1	*Lb. helveticus* DPC 4571 PrtM	99%
3	*Lb. helveticus* H9 PrtP	99%
5	*Lb. helveticus* CNRZ32 OppB1	99%
6	*Lb. helveticus* CNRZ32 OppF1	99%
7	*Lb. helveticus* CNRZ32 OppD1	99%
9	*Lb. helveticus* DPC4571 Pep F	99%
10	*Lb. helveticus* H9 PepO2	99%
11	*Lb. helveticus* DPC 4571 PepE	99%
12	*Lb. helveticus* DPC 4571 PepC	99%
16	*Lb. helveticus* DPC 4571 PepN	99%
17	*Lb. helveticus* D75 PepV	99%
18	*Lb. helveticus* D75 PepA	99%
19	*Lb. helveticus* D75 PepT	99%
20	*Lb. helveticus* D75 PepQ	99%
21	*Lb. helveticus* DPC457 carboxypeptidase	99%
22	*Lb. helveticus* CNRZ32 PepR	99%
23	*Lb. helveticus* CNRZ32 PrtH	99%

## Data Availability

Data are contained within the article or Supplementary Material.

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
