# Peer review of "The Enzyme Gene Expression of Protein Utilization and Metabolism by Lactobacillus helveticus CICC 22171"

_microorganisms, 2022, doi:10.3390/microorganisms10091724_

Round 1

Reviewer 1 Report

The authors studied the hydrolytic ability of Lactobacillus helveticus to β-casein and the expression of the enzyme gene during protein utilization. The article presents the results of the study of the level of hydrolysis depending on the duration. The amount, type, and molecular weight of peptides formed as a result of protein utilization were analyzed. The analysis of the enzymatic systems responsible for the hydrolysis of various sections of the protein molecule was carried out. I believe that the manuscript is written at a high scientific level, the research was carried out using modern high-tech methods, the results were statistically processed and presented in figures and tables. The manuscript can be published in the journal. There are a few comments.

Line 28  the production of pickled foods such as cheese, yogurt, wine, and  kimchi. Not only pickled, but also for fermented food

Line 33  The structural characteristics of proteolytic system of Lactobacillus were similar to that of Lactococcus lactis.

This phrase needs to be clarified. What specific Lactobacillus strains are you talking about?

Line 158 - Figure 1. (a) was β-casein hydFigure 1. (a) – mistake

Author Response

Dear Editor

Thank you very much for your letter with comments on our manuscript entitled “The enzyme gene expression of protein utilization and metabolism by Lactobacillus helveticus CICC 22171" (microorganisms-1863331). We have revised our manuscript according to reviewer’s comments, and would like to re-submit it for your consideration. We have addressed the comments raised by the reviewers, and the amendments are highlighted in red in the revised manuscript. Point by point responses to the reviewers’ comments are listed below.

We hope that the revised version of the manuscript is now acceptable for publication in your journal.

I am looking forward to hearing from you soon.

Kind regards

Yours sincerely,

Zhao Hongfei and Zhang Bolin

Here we would like to express our sincere gratitude to the reviewers for their constructive and positive comments.

Response to Reviewer 1 Comments

Point 1: Line 28  the production of pickled foods such as cheese, yogurt, wine, and kimchi. Not only pickled, but also for fermented food

Response 1: We have made correction according to the reviewer’s comments (Line 29).

Point 2: Line 33  The structural characteristics of proteolytic system of Lactobacillus were similar to that of Lactococcus lactis.

This phrase needs to be clarified. What specific Lactobacillus strains are you talking about?

Response 2: We have added the information in the revised version (Line 35-37).

Point 3: Line 158 - Figure 1. (a) was β-casein hydFigure 1. (a) – mistake 

Response 3: We are sorry for our careless. We have made correction according to the reviewer’s comments (Line 167).

Reviewer 2 Report

The authors have explored the proteolysis-related gene expression profiles and generated peptides during the cultivation of Lb. helveticus CICC 22171, which was isolated from traditional Chinese cheese by them.

The strain surely possesses high level of proteolytic activity, however, the aim of the present study is unobvious.

I wonder why they have analyzed the strain CICC 22171 only, because the strain is not the type strain of Lb. helveticus, and is not reported to produce any useful functional peptides.

The strain has high proteolytic activity as shown in the submitted manuscript, if so, the strain may be different from “standard” type of Lb. helveticus strains.

If the authors want to compare the proteolytic profiles among LAB strains, such as Lc. lactis, they should select several strains of Lb. helveticus which show similar characteristics to type strain.

In addition, why the authors did not perform the whole genome sequencing of the strain CICC 22171?

I think the submitted manuscript does not satisfy the quality of publish as the original article in the journal “Microorganisms”.

Author Response

Dear Editor

Thank you very much for your letter with comments on our manuscript entitled “The enzyme gene expression of protein utilization and metabolism by Lactobacillus helveticus CICC 22171" (microorganisms-1863331). We have revised our manuscript according to reviewer’s comments, and would like to re-submit it for your consideration. We have addressed the comments raised by the reviewers, and the amendments are highlighted in red in the revised manuscript. Point by point responses to the reviewers’ comments are listed below.

We hope that the revised version of the manuscript is now acceptable for publication in your journal.

I am looking forward to hearing from you soon.

Kind regards

Yours sincerely,

Zhao Hongfei and Zhang Bolin

Here we would like to express our sincere gratitude to the reviewers for their constructive and positive comments.

Response to Reviewer 2 Comments

Point 1: The authors have explored the proteolysis-related gene expression profiles and generated peptides during the cultivation of Lb. helveticus CICC 22171, which was isolated from traditional Chinese cheese by them.

The strain surely possesses high level of proteolytic activity, however, the aim of the present study is unobvious.

Response 1: The objective of this study was to investigate in depth the proteolytic activity of strain CICC 22171 and to observe its ability to produce free amino acids and short peptides, as well as the gene expression levels of the associated proteases during hydrolysis, in order to investigate the ability of this strain as a functional fermentor for fermented dairy products. We have rewritten the description in the revised version (Line 82-84).

Point 2: I wonder why they have analyzed the strain CICC 22171 only, because the strain is not the type strain of Lb. helveticus, and is not reported to produce any useful functional peptides.

Response 2: The strain CICC 22171 used in this study has been proved to have probiotic functions. However, only glycometabolism and energy metabolism have been studied in this strain, and its specific protein utilization is unknown. In addition, there is no data to support the gene regulation of the transport system of strain CICC 22171. Therefore, we further explored the proteolytic ability and enzyme gene expression of CICC 22171. In fact, we have discussed about the production of functional peptides by strain CICC 22171 in another published article, as cited below:

Zhao, H., Zhou, F., Wang, L., Bai, F., Dziugan, P., and Walczak, P. Characterization of a bioactive peptide with cytomodulatory effect released from casein. Eur Food Res Technol 2014, 238, 315-322.

Point 3: The strain has high proteolytic activity as shown in the submitted manuscript, if so, the strain may be different from “standard” type of Lb. helveticus strains. 

Response 3: Regarding Reviewer’s comments, we have added some content about this part in the revised version (Line 284-286, 289-291, 294-297, 306-308, 376-384).

Point 4: If the authors want to compare the proteolytic profiles among LAB strains, such as Lc. lactis, they should select several strains of Lb. helveticus which show similar characteristics to type strain.

Response 4: We have added these contents in the revised version according to the comments of reviewers (Line 291-293, 312-314, 332-335, 353-358, 361-365).

Point 5: In addition, why the authors did not perform the whole genome sequencing of the strain CICC 22171?

Response 5: The data of whole genome sequencing of strain CICC 22171 have been published in another article.

Xu, M., Hu, S., Wang, Y., Wang, T., Dziugan, P., Zhang, B., & Zhao, H. Integrated transcriptome and proteome analyses reveal protein metabolism in Lactobacillus helveticus CICC22171. Front Microbiol 2021, 12.

Round 2

Reviewer 2 Report

The revised manuscript seems to meet a basic level of quality for publishing in the journal.

There is a phrase bothering me at line 71–72, “probiotic functions”. As the authors wrote in the text, only glycometabolism and energy metabolism profiles have been studied in the strain, in other words, predicted probiotic functions in the strain were not confirmed by animal experiments or clinical trials. Therefore, the phrase “probiotic functions” should be changed to “potent probiotic functions”.